# The Effect of Common Viral Inactivation Techniques on 16S rRNA Amplicon-Based Analysis of the Gut Microbiota

**DOI:** 10.3390/microorganisms9081755

**Published:** 2021-08-17

**Authors:** Zachary McAdams, Kevin Gustafson, Aaron Ericsson

**Affiliations:** 1Molecular Pathogenesis and Therapeutics Program, University of Missouri, Columbia, MO 65211, USA; zlmg2b@umsystem.edu; 2Department of Veterinary Pathobiology, University of Missouri, Columbia, MO 65211, USA; klgbkt@missouri.edu; 3Comparative Medicine Program, University of Missouri, Columbia, MO 65211, USA; 4Metagenomics Center, University of Missouri, Columbia, MO 65211, USA; 5Mutant Mouse Resource and Research Center, University of Missouri, Columbia, MO 65211, USA

**Keywords:** 16S rRNA 1, viral inactivation 2, gut microbiome (GM) 3, fecal DNA extraction 4, SDS 5, TRIzol 6, Buffer AVL 7

## Abstract

Research investigating the gut microbiome (GM) during a viral infection may necessitate inactivation of the fecal viral load. Here, we assess how common viral inactivation techniques affect 16S rRNA-based analysis of the gut microbiome. Five common viral inactivation methods were applied to cross-matched fecal samples from sixteen female CD-1 mice of the same GM background prior to fecal DNA extraction. The V4 region of the 16S rRNA gene was amplified and sequenced from extracted DNA. Treatment-dependent effects on DNA yield, genus-level taxonomic abundance, and alpha and beta diversity metrics were assessed. A sodium dodecyl sulfate (SDS)-based inactivation method and Holder pasteurization had no effect on measures of microbial richness, while two Buffer AVL-based inactivation methods resulted in a decrease in detected richness. SDS inactivation, Holder pasteurization, and the AVL-based inactivation methods had no effect on measures of alpha diversity within samples or beta diversity between samples. Fecal DNA extracted with TRIzol-treated samples failed to amplify and sequence, making it unsuitable for microbiome analysis. These results provide guidance in the 16S rRNA microbiome analysis of fecal samples requiring viral inactivation.

## 1. Introduction

The gut microbiome (GM) is the diverse community of microorganisms contained within the animal gastrointestinal tract and plays a critical role in host health and physiology [1]. The composition of the GM is dynamic and changes in response to a multitude of factors, including viral infection [2]. Studying the GM during a viral infection is of interest as it may provide further insight into disease progression, disease outcomes, or identification of potential therapeutics. To analyze the GM, the hypervariable regions of the 16S rRNA gene are often amplified from extracted fecal DNA and sequenced; however, handling fecal samples from infected hosts poses a potential risk to the researcher as the samples may harbor infectious viral particles [3,4]. To limit safety risks and comply with appropriate biosafety regulations, the fecal viral load must be inactivated prior to processing and DNA extraction. To our knowledge, the effect of different viral inactivation methods on fecal DNA extraction and targeted amplicon sequencing for GM analysis has not been reported.

Viral inactivation methods can be categorized into three major strategies: pasteurization, detergents, or chemical denaturants. Thermal inactivation is an effective non-invasive technique as it does not require the addition of any reagents [5,6]. In the context of DNA extraction, thermal inactivation methods like Holder pasteurization (63 °C for 30 min) are ideal, as samples are heated at a temperature sufficient for the inactivation of many viral species but remain low enough to maintain DNA integrity for targeted sequence amplification [7,8].

Detergent-based viral inactivation methods frequently utilize sodium dodecyl sulfate (SDS), an anionic detergent that disrupts the membrane of the viral envelope [9,10]. SDS concentrations of 1% or greater have been shown to be effective at inactivating several viral species [9,10]. Moreover, fecal DNA extraction protocols often utilize SDS as a detergent for bacterial cell lysis [11].

Chemical denaturants employ guanidinium thiocyanate as a protein denaturant rendering viruses non-infectious [10,12]. TRIzol Reagent (ThermoFisher Scientific, Waltham, MA, USA) and Buffer AVL (Qiagen, Venlo, the Netherlands) are commercially available nucleic acid extraction reagents that utilize the chaotropic salt as an inactivating agent and are known to be broadly effective against many viral species [10,12]. In addition to guanidinium thiocyanate, TRIzol utilizes phenol as an inactivating agent. Buffer AVL alone is not sufficient to inactivate some viruses and requires an additional pasteurization step to be effective [13].

Here, we simulate viral inactivation in cross-matched fecal samples by applying five common inactivation techniques or reagents—Holder Pasteurization, a SDS lysis buffer, TRIzol Reagent, Buffer AVL, and Buffer AVL with pasteurization—to fecal DNA extraction protocols for targeted 16S rRNA amplification and sequencing. The objective of this work is to assess the treatment-dependent effects of the selected inactivation techniques on DNA yields, measured alpha and beta diversities, and relative taxonomic abundances to guide future 16S rRNA amplicon-based analysis of the GM in the context of a viral infection.

## 2. Materials and Methods

### 2.1. Sample Collection and Processing

Freshly evacuated fecal pellets from 16 female CD-1 mice with an Envigo-origin GM (University of Missouri Mutant Mouse Resource & Research Center, Columbia, MO, USA) [14] were collected simultaneously to minimize temporal changes in the GM. Fecal sample collection was performed by placing a single mouse into an empty autoclaved cage and allowing the mouse to defecate 2–4 fecal pellets. The mouse was then removed, and autoclaved toothpicks were used to remove the fecal pellets. Following collection, fecal pellets were promptly stored at −80 °C until processing. Pellets from each mouse were split into six sections. Each portion was weighed and placed into separate sterile 2 mL round-bottom tubes. Fecal samples were then stored at −20 °C until viral inactivation was simulated, and DNA extracted.

### 2.2. Viral Inactivation Simulation and DNA Extraction

The Qiagen QIAamp PowerFecal Kit (Qiagen, Venlo, The Netherlands) protocol was modified so that samples were mechanically homogenized using a Tissue Lyser II (Qiagen, Venlo, The Netherlands) at 30 Hz for 10 min in 2 mL round-bottom tubes containing a single 0.5 cm stainless steel bead unless otherwise noted.

#### 2.2.1. Control Extractions

DNA from control samples was extracted using the modified Qiagen QIAamp PowerFecal Kit as above.

#### 2.2.2. SDS

For SDS-treated samples, DNA was extracted using the ammonium acetate and isopropanol-based method described by Ericsson et al. [11], which utilizes a 4% SDS lysis buffer. Briefly, fecal samples were mechanically homogenized in 800 μL of lysis buffer (500 mM NaCl, 50 mM Tris-HCl, 50 mM EDTA, and 4% SDS) with a TissueLyser II at 30 Hz for 3 min in 2 mL round-bottom tubes with a single 0.5 cm stainless steel bead. Following a twenty-minute incubation at 70 °C, samples were centrifuged at 5000× *g* for 5 min. The supernatant was transferred to a fresh tube and 200 μL 10 mM ammonium acetate were added and mixed by inversion. The samples were incubated on ice for 5 min then centrifuged at 5000× *g* for 5 min. After transferring the supernatant to a fresh tube, an equal volume of cold isopropanol was added. Samples were then incubated on ice for 30 min before centrifugation at 16,000× *g* at 4 °C for 15 min. The supernatant was discarded, and DNA pellets were washed with 70% ethanol. DNA was resuspended in 150 μL Tris-EDTA (10 mM Tris and 1 mM EDTA) to which 15 μL Proteinase K and 200 μL AL buffer from a DNeasy kit (Qiagen, Venlo, The Netherlands) were added. Samples were incubated at 70 °C for 10 min before 200 μL 100% ethanol were added. The sample was transferred to a DNeasy kit spin column and purified following manufacturer instructions.

#### 2.2.3. Holder Pasteurization

Samples were briefly centrifuged to collect the pellets to the tube bottom then incubated at 63 °C for 30 min [15]. DNA was extracted from the pasteurized pellet using the modified Qiagen QIAamp PowerFecal Kit as above.

#### 2.2.4. TRIzol

DNA extractions followed the manufacturer provided TRIzol Reagent (ThermoFisher Scientific, Waltham, MA, USA) Tissue DNA extraction protocol. Briefly, samples were homogenized for 10 min at 30 Hz with a TissueLyser II in 1 mL TRIzol Reagent. Samples were centrifuged at 13,000× *g* for 5 min and the supernatant was transferred to a fresh tube to remove the remaining debris. Samples were incubated at room temperature for 5 min before 200 μL chloroform were added and mixed by vortexing. Samples were incubated at room temperature for 3 min before centrifugation at 12,000× *g* at 4 °C for 15 min. The clear aqueous layer was discarded and 300 μL absolute ethanol were added. Samples were inverted to mix and incubated for 3 min at room temperature. The DNA was pelleted at 2000× *g* for 5 min at 4 °C before discarding the supernatant. The pellet was washed with 1 mL of 100 mM sodium citrate in 10% ethanol (pH 8.5) and incubated at room temperature for 30 min. The DNA was again pelleted and washed with the sodium citrate solution. The DNA was again pelleted, supernatant discarded, then resuspended in 1.5 mL of 75% ethanol. The DNA was pelleted once more, supernatant discarded, then allowed to air dry for 10 min. The DNA pellet was resuspended in 8 mM NaOH by gently pipetting. The sample was centrifuged at 12,000× *g* at 4 °C for 10 min before transferring the supernatant to a fresh tube.

#### 2.2.5. Buffer AVL

DNA was extracted using a modified Qiagen QIAamp PowerFecal Kit protocol. Briefly, samples were mechanically homogenized in 750 μL Buffer AVL (Qiagen, Venlo, The Netherlands) in place of the kit provided PowerBead Solution. The remainder of the extraction protocol followed the modified Qiagen QIAamp PowerFecal Kit protocol as above.

#### 2.2.6. Buffer AVL with Pasteurization

DNA was extracted using a modified Qiagen PowerFecal Kit protocol. Samples were briefly vortexed in 750 μL Buffer AVL before incubating at 60 °C for 15 min [13]. Samples were homogenized as above following pasteurization. The remainder of the extraction protocol followed the modified Qiagen QIAamp PowerFecal Kit protocol as above.

#### 2.2.7. 16S rRNA Amplification and Sequencing

DNA concentrations of the fecal lysate and purified yields were quantified fluorometrically (Qubit dsDNA BR Assay, Life Technologies, Carlsbad, CA, USA). Samples with concentrations below the limit of detection (LOD, 0.010 ng/mL) were recorded as one half of the LOD. Samples were diluted to 3.51 ng/μL before 16S rRNA amplification. Samples with a concentration lower than 3.51 ng/μL were concentrated to approximately 60 μL under vacuum centrifugation at 37 °C. A 60 μL aliquot of DNA was submitted for amplification and sequencing for each sample.

The V4 hypervariable region of the 16S rRNA gene was amplified with the dual-indexed universal primers (U515F/806R) [16] and flanking Illumina adapters. Each polymerase chain reaction (PCR) contained extracted DNA (up to 100 ng), U515F/806R universal primers (0.2 µM), dNTPs (200 µM), and high-fidelity DNA polymerase (Phusion, 1U). The PCR amplification protocol was as follows: 98 °C (3:00) + (98 °C (0:15) + 50 °C (0:30) + 72 °C (0:30)) × 25 cycles + 72 °C (7:00) [11]. The 16S rRNA amplicon libraries were pooled and purified with Axygen Axyprep MagPCR clean-up beads. Purified amplicons were diluted to the appropriate concentration for sequencing with an Illumina MiSeq platform using V2 chemistry. 2 × 250 bp reads were generated from sequencing.

### 2.3. Informatics

Informatics were performed using QIIME2 v2021.2 [17]. Demultiplexed paired-end reads were trimmed of the universal primers and Illumina adapters using Cutadapt [18]. The trimmed sequences were denoised into amplicon sequence variants (ASVs) with DADA2 and phylogeny was determined using a de novo Mafft FastTree approach [19,20,21]. Feature tables containing the frequency of unique ASVs observed in each sample were rarefied to 40,000 total features to maximize the proportion of subsampled features (i.e., distinct ASVs) while minimizing the number of discarded samples for diversity analyses [22,23]. Taxonomy for each ASV was assigned with a sklearn feature classifier algorithm using the *readytowear* 16S rRNA 515F-806R SILVA 138 reference database weighted by “animal distal gut” microbial abundances [24,25,26,27]. All informatics code can be accessed at https://github.com/ericsson-lab/viral-inactivation.git (accessed on 6 July 2021).

### 2.4. Statistical Analysis

Alpha diversity metrics, one-way permutational analysis of variance (PERMANOVA) tests using Bray–Curtis and Jaccard similarities, fourth-root transformations to normalize data, and principal coordinate analyses (PCoA) were generated with the open-access Past 4.04 software [28]. One-way ANOVA and Tukey HSD post hoc tests were performed with the open-access statistical softwares R v3.6.2 or MetaboAnalyst 5.0 [29,30], with a *p* value of 0.05 or less considered statistically significant. Three-dimensional PCoAs using Bray–Curtis similarities were generated using the EMPeror plug-in [31] within QIIME2 v2021.2 and visualized at view.qiime2.org (accessed on 6 July 2021).

## 3. Results

### 3.1. Not All Viral Inactivation Methods Yield Sequencing-Quality DNA

DNA concentrations of the fecal lysate and purified yields for each treatment were normalized to the sample mass (Table 1). Significant differences in the fecal lysate DNA concentration between the control extractions and AVL (*p* = 7.8 × 10^−3^), AVL + Heat (*p* = 0.010), and TRIzol (*p* = 5.5 × 10^−6^) groups were observed (Appendix A). Successful 16S rRNA amplification and sequencing were considered to be samples that yielded a per sample sequencing read count greater than 10,000, as this sequencing depth is sufficient (with similar samples) to detect most rare taxa [11]. Samples yielding less than 10,000 reads were excluded from downstream analysis. While TRIzol-treated samples did yield DNA at concentrations above the LOD, these samples did not produce any successful reads (Table 1). DNA collected from AVL and AVL + Heat inactivation methods frequently yielded DNA concentrations below the LOD but produced completely successful 16S rRNA sequencing data (Table 1). All other samples except one within the control group produced successful read counts. Every TRIzol sample was omitted from downstream analyses as no sample produced successful 16S sequencing results. Four additional samples were removed from subsequent analyses after rarefication (Table 1).

### 3.2. Treatment-Dependent Effects on Microbial Richness but Not Distribution or Beta Diversity

Data were assessed for treatment-dependent effects on alpha diversity measures of richness and distribution. No significant differences were detected in the observed or predicted richness between samples in the SDS and Holder pasteurization groups comparted to the control (Figure 1A,B); however, significant differences in both the observed (*p* = 2.0 × 10^−3^) and predicted (*p* = 7 × 10^−4^) richness were detected in samples subjected to AVL inactivation. A significant difference in the predicted richness between AVL + Heat inactivation and control was observed (*p* = 0.021). No significant differences in the distribution of features within samples using Simpson or Shannon indices were observed between inactivation methods (Figure 1C,D).

Next, the relative abundance of taxa at the genus level within each sample from individual donors was compared to subjectively assess the effect of inactivation method on detected taxonomy (Figure 2). Only 5 of the 120 detected genera were found to be significantly different between inactivation methods, suggesting little effect of the inactivation method on determined taxonomy (Appendix A). To objectively assess the effect of inactivation on beta diversity between samples, two-dimensional PCoAs of the rarefied feature table with a fourth-root transformation to normalize data showed a high similarity between all inactivation methods using both weighted (Bray–Curtis, *p* = 0.21, F = 1.172) and unweighted (Jaccard, *p* = 0.39, F = 1.019) metrics (Figure 3). Pairwise comparisons revealed a significant, albeit subtle, difference between the AVL and Holder inactivation groups using Jaccard similarities (*p* = 0.044, F = 1.609).

Consistent with the two-dimensional PCoAs, three-dimensional PCoA plots showed no clustering of samples when labeled by inactivation method but did demonstrate clustering when samples were labeled by donor using both weighted (Appendix A) and unweighted metrics (Appendix A). Interactive three-dimensional PCoAs are available for download at https://github.com/ericsson-lab/viral-inactivation.git (accessed on 6 July 2021) and are viewable at view qiime2.org (accessed on 6 July 2021).

## 4. Discussion

While several studies have tested the efficacy of these and other inactivation methods against numerous viruses, the current study was designed to assess the influence of several methods of viral inactivation on the suitability of residual DNA in the fecal matrix for use as templates in downstream targeted amplicon sequencing. These methods were selected based on feasibility for a researcher working in high BSL conditions to inactivate the fecal viral load before transporting samples to a facility of the appropriate biosafety level for DNA extraction, and for their likelihood of preserving DNA integrity of the samples. These methods are not effective against all virus types, or in all sample matrices, making it important for researchers to validate that their virus of interest can be inactivated in fecal biomass by one of these methods before DNA is extracted for 16S rRNA library generation and sequencing.

Our data indicate that the use of a surfactant (SDS)-based lysis buffer or heat-based inactivation (Holder pasteurization) have no detectable effect on final assessments of alpha or beta diversity, making them preferable viral inactivation techniques for samples intended for 16S rRNA analysis, assuming they are effective against the virus in question. Guanidinium-based buffers (i.e., TRIzol and AVL) produced mixed results. While TRIzol is highly effective in the inactivation of many viruses within the Alphavirus, Bunyavirus, Filovirus, Flavivirus, Simplexvirus, Adenovirus, Enterovirus, Influenzavirus, and Coronavirus (e.g., SARS-CoV2) genera [10,12,32], it is not supported by the current data as a means of viral inactivation of samples intended for targeted amplicon sequencing. Fecal DNA extracted with TRIzol may require further purification to remove residual amplification or sequencing inhibitors, despite detectable, albeit reduced, DNA yields.

It is unclear why the methods including AVL (either with or without heat) resulted in such low DNA yields. It is also notable that, despite those low yields, the AVL-inactivated samples were still able to provide meaningful data regarding beta diversity, although alpha diversity was negatively affected. It is unknown however whether the observed differences were due to a sequence bias exhibited by the buffer during extraction or if rarefying a greater proportion of total features favored lower measured richness, as rare taxa may not be represented (Table 1). Regardless, these data would suggest that, while not optimal, AVL-inactivated fecal samples could still yield meaningful targeted amplicon sequencing data, assuming all samples were inactivated in an identical manner. Like TRIzol, it is possible that AVL-based methods could be optimized with additional purification steps. Regardless, when considering DNA yields and sequencing data collectively, SDS- and heat-based inactivation protocols are preferable platforms for fecal samples intended for targeted amplicon sequencing. Collectively, these data will help guide the selection of a method for fecal viral inactivation for 16S rRNA amplicon-based microbiome analysis.

## Figures and Tables

**Figure 1 microorganisms-09-01755-f001:**
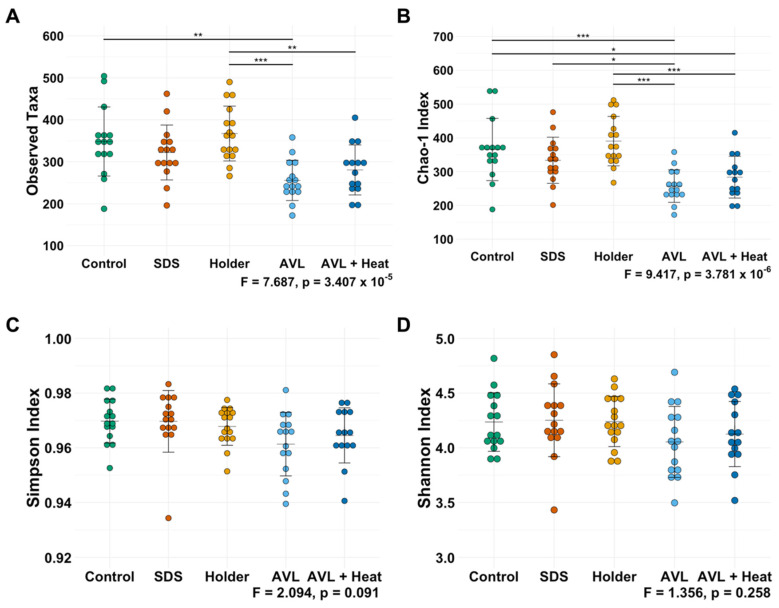
Alpha diversity metrics per viral inactivation method: (**A**) Observed Taxa, (**B**) Chao-1 Index, (**C**) Simpson Index, (**D**) Shannon Index; Control (*n* = 15), SDS (*n* = 15), TRIzol (omitted), Holder (*n* = 16), AVL (*n* = 15), AVL + Heat (*n* = 14). Dots represent individual data points, bars represent mean ± SD, one-way ANOVA followed by Tukey HSD post-hoc test; * *p* < 0.05, ** *p* < 0.01, *** *p* < 0.001.

**Figure 2 microorganisms-09-01755-f002:**
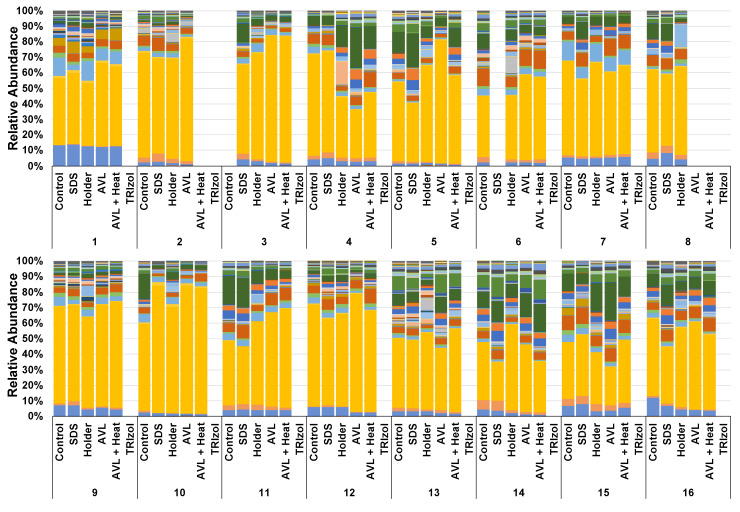
Stacked bar chart displaying per sample relative taxonomic abundance at the genus level. Samples are organized by sample donor. Treatments labels with no bar indicate the samples either did not successfully sequence or did not pass the rarefaction filter.

**Figure 3 microorganisms-09-01755-f003:**
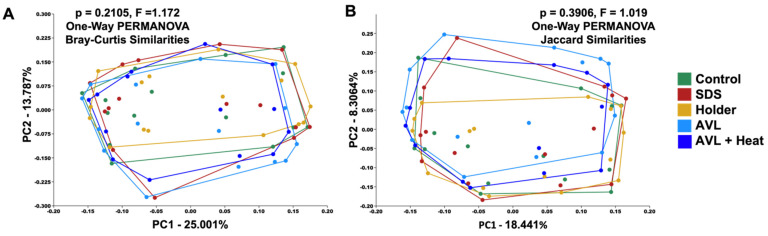
Two-dimensional principal coordinate analysis of (**A**) Bray–Curtis and (**B**) Jaccard similarities; Control (*n* = 15), SDS (*n* = 15), TRIzol (omitted), Holder (*n* = 16), AVL (*n* = 15), AVL + Heat (*n* = 14); one-way PERMANOVA.

**Table 1 microorganisms-09-01755-t001:** Summary of DNA yields, sequencing success, feature count after denoising, and sample number passing the 40,000-feature rarefication filter for each simulated viral inactivation method. Mean ± SD.

Inactivation Method	DNA Yield (ng DNA/mg Feces, *n* = 16)	Successful 16S Sequencing (≥10,000 Reads)	Features per Sample Post-Denoising (Mean ± SD, *n* = 16)	Sample Number Passing 40,000 Feature Rarefication Filter
Control	135.9 ± 106.2	15/16	99,572 ± 34,364	15/16
SDS	242.0 ± 186.0	16/16	84,754 ± 17,681	15/16
TRIzol	18.7 ± 19.3	0/16	0	0/16
Holder	65.3 ± 29.5	16/16	111,909 ± 17,831	16/16
AVL	0.2 ± 0.5	16/16	62,508 ± 18,162	15/16
AVL + Heat	0.4 ± 0.8	16/16	72,796 ± 21,625	14/16

## Data Availability

Sequences can be accessed at the National Center for Biotechnology Information, Sequence Read Archive (SRA) with the BioProject ID PRJNA71299. All code used to process 16S rRNA sequences and interactive PCoA plots can be accessed at: https://github.com/ericsson-lab/viral-inactivation.git (accessed on 6 July 2021).

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
