# Peer review of "The Effect of Common Viral Inactivation Techniques on 16S rRNA Amplicon-Based Analysis of the Gut Microbiota"

_microorganisms, 2021, doi:10.3390/microorganisms9081755_

Round 1

Reviewer 1 Report

  1. Did you test the efficacy of the DNA extraction method without the additional viral inactivation to determine viral inactivation by the DNA methodology alone? The purpose of the study was if you have a higher BSL virus and want to move the samples to a lower BSL to conduct the DNA extraction, but should it be incorporated as a precautionary method? What is the utility outside of the described use-case?

  1. Did you spike any of the samples with virus to ensure virus inactivation with these protocols in this matrix?

  1. In the discussion, line 238 you mention you used this DNA extraction method since it is widely used, did you try any other widely used techniques, like those used in the HMP?

  1. can you expand on this statement starting on line 235” These methods are not effective against all virus types making it important to ensure the virus of interest can be inactivated in fecal biomass by one of these methods before DNA extraction for 16S rRNA amplicon-based analysis of the gut microbiome.” For the purposes of this study, does it matter if you get good microbiome data if the virus is not inactivated?

  1. Were there any instances in which the extracted DNA samples with the additional viral inactivation steps yielded superior DNA compared with the control? From you data, it appears as though the best pure yield was the SDS or control (large SD on those two though), but it appears as though the Holder methodology had superior features per sample (Table 1), and was also the only one that passed the rarefication Filter, can you elaborate on some of the trends that you saw? What lead to the high deviations?

  1. At the genus level you said you subjectively assessed the effect (line 213), how? Did you notice any consistent genus level differences, suggesting certain extraction methodologies were favoring specific microbes? More info on the assessment needed.

Author Response

Response to Reviewer 1 Comments

Point 1: Did you test the efficacy of the DNA extraction method without the additional viral inactivation to determine viral inactivation by the DNA methodology alone? The purpose of the study was if you have a higher BSL virus and want to move the samples to a lower BSL to conduct the DNA extraction, but should it be incorporated as a precautionary method? What is the utility outside of the described use case?

The goal of this study was not to determine the efficacy of viral inactivation techniques as none of the selected methods are effective against all viral species, but rather how selected methods would affect downstream 16S rRNA sequencing. We applied known viral inactivation methods to DNA extraction from fecal biomass so that future investigators may utilize a method effective against their viral species of interest knowing how that method will affect 16S rRNA amplicon-based microbiome analysis.

Samples collected at a high BSL level must be rendered safe for transport into a lower BSL level facility. The pathogenic agents within these samples may not be viral but would still require an institutionally approved inactivation method to bring the samples out of a high BSL level facility, thus, the methods presented here could be applied outside of the described use-case given institutional biosafety requirements.

Point 2: Did you spike any of the samples with virus to ensure virus inactivation with these protocols in this matrix?

The objective of this work was to access how common viral inactivation techniques affect downstream 16S rRNA amplicon-based analysis of the gut microbiome. We did not spike any of the samples to evaluate virus inactivation efficacy. Verifying inactivation of the fecal viral load was not within the scope of this work.

Point 3: In the discussion, line 238 you mention you used this DNA extraction method since it is widely used, did you try any other widely used techniques, like those used in the HMP?

To our knowledge, the Human Microbiome Project used QIAGEN PowerSoil Kits which utilize an identical chemistry as the QIAGEN PowerFecal Kits utilized in the Control group of this study. The QIAGEN PowerFecal Kit (Control) and Isopropanol-based (SDS) extraction methods are well-established protocols for fecal DNA extraction.

Point 4: Can you expand on this statement starting on line 235” These methods are not effective against all virus types making it important to ensure the virus of interest can be inactivated in fecal biomass by one of these methods before DNA extraction for 16S rRNA amplicon-based analysis of the gut microbiome.” For the purposes of this study, does it matter if you get good microbiome data if the virus is not inactivated?

The objective of this work was to assess how common viral inactivation methods would affect downstream 16S rRNA sequencing. We applied known viral inactivation methods to DNA extraction from fecal biomass so that future investigators may utilize a method effective against their viral species of interest knowing how that method will affect 16S rRNA amplicon-based microbiome analysis. If the inactivation method is not effective against a researcher’s viral species of interest, then those samples could not be transported to a lower BSL facility for DNA extraction and 16S rRNA amplicon-based microbiome analysis. It is not necessarily important, for the purposes of this study, as to whether or not the tested methods work with any specific virus, and our statement was intended to convey that the method of viral inactivation selected by a researcher should be governed by both the data we present as well as virus-specific testing to ensure proper lab biosafety and biocontainment.

Point 5: Were there any instances in which the extracted DNA samples with the additional viral inactivation steps yielded superior DNA compared with the control? From your data, it appears as though the best pure yield was the SDS or control (large SD on those two though), but it appears as though the Holder methodology had superior features per sample (Table 1), and was also the only one that passed the rarefication Filter, can you elaborate on some of the trends that you saw? What lead to the high deviations?

The Reviewer raises an excellent point. Fecal DNA extractions produced variable DNA yields for many reasons including input sample mass, user technique, or unknown properties inherent to the sample, thus, DNA yields are not an optimal determinant of sequencing success, as our lab has seen many times. Low-yield samples can produce high-coverage data, and vice versa depending on many factors. Primer bias during 16S rRNA amplification or the competition between the pooled amplicon libraries binding to the flow cell during sequencing may also contribute to the variability seen in 16S rRNA sequencing data [1].

To minimize some of the variability that may arise during DNA extraction or 16S rRNA amplification and sequencing, the number of detected features per sample were rarefied to an equal number. It was necessary to exclude the samples that did not pass the rarefication filter to equally compare the effect of the inactivation method on 16S rRNA gut microbiome analysis. The rarefication threshold was chosen to maximize the number of randomly selected features and minimize the number of excluded samples, mimicking the real-world scenario wherein users would want to optimize the balance between number of samples included and coverage. This threshold was subjectively chosen based on alpha rarefaction curves and is not indicative of the treatment-dependent effects.

Point 6: At the genus level you said you subjectively assessed the effect (line 213), how? Did you notice any consistent genus-level differences, suggesting certain extraction methodologies were favoring specific microbes? More info on the assessment needed.

We appreciate the Reviewer’s suggestion and performed additional statistical tests accordingly. Significant differences in genus-level relative abundance between inactivation methods were assessed using a one-way analysis of variance and Tukey-HSD post hoc tests. Significant differences were considered to be a p value of less than 0.05. The post hoc analysis found only 5 of the 120 detected genera to be significantly different between inactivation methods. Of these 5 significantly different taxa, no trends were observed regarding the taxonomic relatedness of taxa detected at different levels between treatment groups. These findings have been added to the Results section and supplementary files as Supplementary Table 1 (See lines 215-218, Supplementary Table 1).

Reference

  1. Morgan, X.C.; Huttenhower, C. Meta’omic Analytic Techniques for Studying the Intestinal Microbiome. Gastroenterology 2014, 146, 1437-1448.e1, doi:10.1053/j.gastro.2014.01.049.

Reviewer 2 Report

The manuscript by McAdams and colleagues explores the effect of viral inactivation methods on the downstream 16S PCR amplification and NGS sequencing for gut microbiome analyses.

The manuscript is well written and deals with a topic of great interest for researchers that study -especially human- microbiota.

Results showed that most procedures are suitable (but with some, poorly explained and expected yields differences), except for that using Trizol.

However, some of the extraction methods employed, as well and the selected inactivation protocols raises some doubts and concerns.

As for the extraction methods, some protocols, like that using Buffer AVL, are adapted (and tested?) procedures that seem to not ensure the maximum expected effectiveness. In this regard, probably the authors could have considered first the suitability of the procedures for the specific use (i.e fecal DNA extraction).

Also, the choice of Trizol is quite surprising. Indeed, it is undoubtedly effective in inactivating nearly any biological agent, but it is primarily intended for RNA extraction (although, protocols for simultaneous RNA, proteins, and DNA extraction are proposed, as the authors correctly report). Moreover, Trizol is not an agent usually used for fecal DNA extraction (for fecal RNA extraction is reported as effective, instead). Furthermore, the Trizol action relies on the simultaneous effect of GuSCN and phenol (authors should include such details in the introduction), which probably is poorly compatible with feces.  

In my opinion, an effective inactivating solution (which might overcome the partial ineffectiveness of Buffer AVL) to be tested could have been (and might be) the original formulation proposed by Chomczynski, 1993 (Trizol is based on that), where a proper GuSCN concentration (4M) is combined with an anionic surfactant, which may result in viral inactivation without other additions. This formula, as well as similar ones (with even higher GuSCN concentrations, exceeding 5M), allows for DNA extraction, provided that the acidification step is omitted. DNA can be recovered by precipitation or by silica binding (both with or without organic extraction). 

Among other alternative procedures that could have been tested, CTAB-based extraction methods might deserve attention.

Why the authors have included Trizol, not considering other (probably more suitable and effective) GuSCN-based protocols or CTAB protocols?

Including such methods (some of which are employed for fecal DNA extraction) in the comparison of protocols would greatly improve the manuscript.

Author Response

Response to Reviewer 2 Comments

The manuscript by McAdams and colleagues explores the effect of viral inactivation methods on the downstream 16S PCR amplification and NGS sequencing for gut microbiome analyses. The manuscript is well written and deals with a topic of great interest for researchers that study -especially human- microbiota.

Results showed that most procedures are suitable (but with some, poorly explained and expected yields differences), except for that using Trizol. However, some of the extraction methods employed, as well and the selected inactivation protocols raises some doubts and concerns.

Point 1: As for the extraction methods, some protocols, like that using Buffer AVL, are adapted (and tested?) procedures that seem to not ensure the maximum expected effectiveness. In this regard, probably the authors could have considered first the suitability of the procedures for the specific use (i.e fecal DNA extraction).

The Reviewer raises an excellent point. The methods utilized in this work were selected based on their prevalence in the literature describing viral inactivation and our additional criteria including minimal reagents or equipment required for inactivation, and the practicality of implementing the inactivation method in a BSL 3, or higher, facility. While the DNA yields from the Buffer AVL-based methods were low, every sample produced greater than 10,000 sequencing reads - the threshold for a successful sequencing reaction as defined by our lab [1]. Additionally, our lab frequently extracts DNA from low biomass samples with undetectable DNA yields that consistently produce successful sequencing runs, thus, poor DNA yields do not always indicate or predict poor sequencing results.

Point 2: Also, the choice of Trizol is quite surprising. Indeed, it is undoubtedly effective in inactivating nearly any biological agent, but it is primarily intended for RNA extraction (although, protocols for simultaneous RNA, proteins, and DNA extraction are proposed, as the authors correctly report). Moreover, Trizol is not an agent usually used for fecal DNA extraction (for fecal RNA extraction is reported as effective, instead). Furthermore, the Trizol action relies on the simultaneous effect of GuSCN and phenol (authors should include such details in the introduction), which probably is poorly compatible with feces.  

The Reviewer again makes an excellent point. TRIzol was selected based on its prevalence in the literature describing viral inactivation, the commonness of the reagent in laboratories, and the practicality of applying the reagent in a BSL 3, or higher, facility. Because TRIzol is a common reagent in the laboratory, it could easily be applied to fecal biomass. As the Reviewer has suggested, TRIzol is probably incompatible for DNA extraction in fecal biomass which is important to report as the reagent is frequently utilized for viral inactivation in other media. These results may deter future investigators from utilizing the reagent for fecal DNA extraction when viral inactivation is required.

The detail regarding the use of GuSCN and phenol in TRIzol has been included in the Introduction (see lines 58-59).

Point 3: In my opinion, an effective inactivating solution (which might overcome the partial ineffectiveness of Buffer AVL) to be tested could have been (and might be) the original formulation proposed by Chomczynski, 1993 (Trizol is based on that), where a proper GuSCN concentration (4M) is combined with an anionic surfactant, which may result in viral inactivation without other additions. This formula, as well as similar ones (with even higher GuSCN concentrations, exceeding 5M), allows for DNA extraction, provided that the acidification step is omitted. DNA can be recovered by precipitation or by silica binding (both with or without organic extraction). 

The Reviewer again raises an excellent point for future consideration. While TRIzol is a GuSCN and phenol-based nucleic acid extraction reagent, it is probably incompatible with DNA extraction from fecal biomass. The Reviewer proposed GuSCN-based methods that, while out of the scope of the current work, may improve fecal DNA yields. The primary challenge with performing additional treatment groups is the lack of remaining samples from the same animals tested in the current work, needed for a controlled comparison.  To add additional treatment groups would require repeating the entire experiment, which is not feasible at this time.

Point 4: Among other alternative procedures that could have been tested, CTAB-based extraction methods might deserve attention. Why the authors have included Trizol, not considering other (probably more suitable and effective) GuSCN-based protocols or CTAB protocols? Including such methods (some of which are employed for fecal DNA extraction) in the comparison of protocols would greatly improve the manuscript.

While we appreciate this recommendation, the logistics of applying a CTAB-based fecal DNA extraction method in a BSL 3 or higher facility are not within the scope of our work. The reported CTAB-based fecal DNA extraction protocol requires extensive sample processing before CTAB - the proposed viral inactivation agent – is applied [2]. The methods selected for this study were chosen based on the ubiquity of materials in a BSL 3 facility and the amount of effort required for inactivation. While the reagents used in the CTAB fecal DNA extraction method may be commonplace in a laboratory, we felt that the effort required to implement the CTAB fecal DNA extraction protocol under BSL 3, or higher, conditions would make this method impractical for 16S rRNA amplicon-based analysis of the microbiome.

References

  1. Ericsson, A.C.; Davis, J.W.; Spollen, W.; Bivens, N.; Givan, S.; Hagan, C.E.; McIntosh, M.; Franklin, C.L. Effects of Vendor and Genetic Background on the Composition of the Fecal Microbiota of Inbred Mice. Plos One 2015, 10, e0116704, doi:10.1371/journal.pone.0116704.
  2. Zhang, B.-W.; Li, M.; Ma, L.-C.; Wei, F.-W. A Widely Applicable Protocol for DNA Isolation from Fecal Samples. Biochem Genet 2006, 44, 494, doi:10.1007/s10528-006-9050-1.

Round 2

Reviewer 1 Report

The purpose of the paper is to determine downstream effects of viral inactivation on subsequent 16S sequencing. While valuable, the paper would be stronger if you either mention the uses of each of the chosen methods (viral type etc), with citations showing inactivation in your matrix (vs breast milk or others) when processed according to your protocols, or providing supplemental data showing each process that preserved the 16S also had the desired intent of inactivation. 

Author Response

Response to Reviewer 1 Comments

Reviewer 1

The purpose of the paper is to determine downstream effects of viral inactivation on subsequent 16S sequencing. While valuable, the paper would be stronger if you either mention the uses of each of the chosen methods (viral type etc), with citations showing inactivation in your matrix (vs breast milk or others) when processed according to your protocols, or providing supplemental data showing each process that preserved the 16S also had the desired intent of inactivation. 

We understand the Reviewer’s concern and apologize for not addressing it adequately in our initial response. While we would still respectfully maintain that testing the various inactivation methods to confirm that they are effective is beyond the scope of the study, we also appreciate that addition of such information would increase the value of the manuscript to readers working in the field. As such, we have followed the Reviewer’s recommendation to summarize the efficacy of each of the tested methods against different virus types, and provide relevant supporting citations. We hope that this adequately addresses the Reviewer’s concerns and sincerely appreciate their careful reading of our manuscript and suggestions for improvement.

Reviewer 2 Report

The revised version of the manuscript includes the only theoretical point (the presence of phenol in TRIzol) that was required. While, in their responses the authors, although appreciating the reviewer's comments, actually seem to evade the actual point of the questions.

I well understand the point of view of the authors and my overall opinion about the manuscript is positive. However, I still think that such a paper, given the results, should not be limited to a comparison between commonly used viral inactivation methods (which is very useful, anyway), but should explore solutions, as well. Moreover, it should aim at suggesting the proper inactivation method(s) when downstream NGS analyses have to be performed.

The suitability of the reads achieved has not been questioned. However, the authors well know that very low copy number templates, especially in the context of very complex communities (i.e bearing high alpha-diversity) might affect the robustness of final results (even if the authors seem to have not experienced these problems in their experiments). Thus, as a general rule, DNA extraction methods should aim to the maximum yield, even extraction power among taxa, reproducibility, and quality. The results, instead, raise the question of the actual effectiveness of guanidinium-based buffers (AVL has to be combined with heating, while TRIzol resulted unsuitable for downstream applications). To test a guanidinium-based buffer that could avoid heating (thus simplifying the procedure), ensuring effectiveness in both viral inactivation and DNA extraction is fully fitting with the scope of the paper. It is really difficult to understand why authors think that it would be out of the scope. The present comparison is suitable and useful, of course, but clarifying the "question of guanidine" is needed.

Author Response

Response to Reviewer 2 Comments

Reviewer 2

The revised version of the manuscript includes the only theoretical point (the presence of phenol in TRIzol) that was required. While, in their responses the authors, although appreciating the reviewer's comments, actually seem to evade the actual point of the questions.

I well understand the point of view of the authors and my overall opinion about the manuscript is positive. However, I still think that such a paper, given the results, should not be limited to a comparison between commonly used viral inactivation methods (which is very useful, anyway), but should explore solutions, as well. Moreover, it should aim at suggesting the proper inactivation method(s) when downstream NGS analyses have to be performed.

We regret that our initial responses appeared evasive and assure the Reviewer that this was not our intent. Regarding the recommendation that the present study should “explore solutions” and “aim at suggesting the proper inactivation method(s) when downstream NGS analyses have to be performed,” we have added substantial text to the Discussion, providing a summary of the efficacy of each of the tested methods against various virus types, with supporting citations. We believe this substantially increases the utility of the manuscript as a resource for researchers. As for the possibility of exploring additional methods or adaptations, we have no remaining sample material from the mice tested in the present study, which would be required (i.e., matched samples from the same animals) in order to compare the effects of sample treatment in a controlled manner, and believe that the tested methods still provide novel and useful data to the research community. We hope that this adequately addresses the Reviewer’s concerns.

The suitability of the reads achieved has not been questioned. However, the authors well know that very low copy number templates, especially in the context of very complex communities (i.e bearing high alpha-diversity) might affect the robustness of final results (even if the authors seem to have not experienced these problems in their experiments). Thus, as a general rule, DNA extraction methods should aim to the maximum yield, even extraction power among taxa, reproducibility, and quality. The results, instead, raise the question of the actual effectiveness of guanidinium-based buffers (AVL has to be combined with heating, while TRIzol resulted unsuitable for downstream applications). To test a guanidinium-based buffer that could avoid heating (thus simplifying the procedure), ensuring effectiveness in both viral inactivation and DNA extraction is fully fitting with the scope of the paper. It is really difficult to understand why authors think that it would be out of the scope. The present comparison is suitable and useful, of course, but clarifying the "question of guanidine" is needed.

We understand part of the Reviewer’s questions regarding guanidinium-based methods but are confused regarding the comments about testing a “guanidinium-based buffer that could avoid heating”, as the present study did that with AVL (tested with and without heating).  The similarity in the results from those groups, and disparity of both groups with the Holder pasteurization method, strongly suggest that the AVL buffer results in low DNA yields for unknown reasons and a reduction in detected richness.  To perform any additional studies comparing other methods or adaptation of tested methods would require collecting samples from an entirely new group of mice and repeating the entire study, as outcome measures need to be controlled for the donor animals and we have no residual fecal material from these animals (and limited funds for additional experiments). As such, we have done our best to describe the concerns associated with guanidinium-based inactivation protocols in the Discussion. We have substantially expanded the Discussion section in response to both Reviewers’ comments, and hope that these additions address Reviewer 2’s concerns.